# A Machine-Learning Tool Concurrently Models Single Omics and Phenome Data for Functional Subtyping and Personalized Cancer Medicine

**DOI:** 10.3390/cancers12102811

**Published:** 2020-09-30

**Authors:** Gift Nyamundanda, Katherine Eason, Justin Guinney, Christopher J. Lord, Anguraj Sadanandam

**Affiliations:** 1Division of Molecular Pathology, The Institute of Cancer Research, London SW3 6JB, UK; gift.nyamundanda@icr.ac.uk (G.N.); kate.eason@icr.ac.uk (K.E.); 2Sage Bionetworks, Seattle, WA 98121, USA; justin.guinney@sagebase.org; 3The Breast Cancer Now Toby Robins Research Centre, The Institute of Cancer Research, London SW3 6JB, UK; Chris.lord@icr.ac.uk

**Keywords:** machine learning, breast cancer, continuous subtypes, functional subtypes, CDK inhibitor, etoposide, subtyping, genome-phenome integration, phenotypes, dimension reduction methods

## Abstract

**Simple Summary:**

Tumours are heterogeneous that reflect variable patient prognosis and treatment responses (phenotypes). Since these variable phenotypes are outcomes of genomics, it is essential to integrate genome and phenome jointly. In this study, we report the development and application of a new machine learning tool (Phenotypic Mapping; *PhenMap*) to identify unsupervised clinically-relevant (functional) subtypes and biomarkers by simultaneously integrating clinical phenotypes and single omics data, mainly, transcriptome. This integrative analysis provides the opportunity to simultaneously identify robust and context-specific subtypes and associated phenotypes (that biologically explain each subtype) without statistically losing information. We demonstrate its utility using breast cancer cell lines and patient samples to identify functional subtypes associated with specific drug responses (including CD4/6 inhibitor) and prognosis. These subtypes may potentially predict clinical outcomes with further validation. This tool can be applied to other omics data such as methylomics, genomics and radiomics along with any phenotypic data.

**Abstract:**

One of the major challenges in defining clinically-relevant and less heterogeneous tumor subtypes is assigning biological and/or clinical interpretations to etiological (intrinsic) subtypes. Conventional clustering/subtyping approaches often fail to define such subtypes, as they involve several discrete steps. Here we demonstrate a unique machine-learning method, phenotype mapping (*PhenMap*), which jointly integrates single omics data with phenotypic information using three published breast cancer datasets (n = 2045). The *PhenMap* framework uses a modified factor analysis method that is governed by a key assumption that, features from different omics data types are correlated due to specific “hidden/mapping” variables (context-specific mapping variables (CMV)). These variables can be simultaneously modeled with phenotypic data as covariates to yield functional subtypes and their associated features (e.g., genes) and phenotypes. In one example, we demonstrate the identification and validation of six novel “functional” (discrete) subtypes with differential responses to a cyclin-dependent kinase (CDK)4/6 inhibitor and etoposide by jointly integrating transcriptome profiles with four different drug response data from 37 breast cancer cell lines. These robust subtypes are also present in patient breast tumors with different prognosis. In another example, we modeled patient gene expression profiles and clinical covariates together to identify continuous subtypes with clinical/biological implications. Overall, this genome-phenome machine-learning integration tool, *PhenMap* identifies functional and phenotype-integrated discrete or continuous subtypes with clinical translational potential.

## 1. Introduction

Cancers represent a heterogeneous collection of diseases with different molecular features, prognoses, and responses to treatment [1,2,3,4,5]. Ultimately, personalized cancer medicine seeks to match the most efficient drug or drug combinations to individual tumor characteristics. To achieve this, omics data must be integrated with phenotypic information, i.e., clinicopathological data such as tumor grade, stage, and/or drug responses. The omics data can represent any single large-scale data including, but not limited to, genome, transcriptome, methylome, or large-scale image data (from computerized tomography, ultrasound, or immunohistochemistry (IHC)). This omics-phenotype integration has hitherto been achieved using “conventional” unsupervised approaches such as hierarchical clustering [6] and non-negative matrix factorization (NMF) [3,5,7], which generally proceed through several discrete statistical steps: (a) unsupervised clustering of molecular data to identify subtypes with “unknown” biological/clinical implications; (b) multiple univariate and multivariate statistical analyses with clinicopathological data to reveal biological/clinical associations; and (c) supervised analysis (e.g., with statistical analysis of microarrays [8]) to identify subtype-specific features (e.g., genes). In addition, conventional low dimension reduction methods, such as principal component analysis and (general) factor analysis do not model phenotypic data jointly with omics data to identify clinically relevant or functional groups. These methods also require a multi-step process to associate omics data with phenotypes. However, multi-step approaches reduce the statistical power and lose information to discover robust, clinically relevant groups and their associated phenotypes and molecular features. This conventional subtype discovery process has in the past, led to slow or no translation of etiological subtypes into the clinic for certain cancer types (including breast cancer), due to the lack of statistical power (because of multiple steps are involved) to gauge the clinical utility of the etiological subtypes [9,10]. The best way to identify robust subtypes or groups that can be translated into the clinic for patient benefit should involve concurrent statistical integration of omics data with phenome.

Hence, we propose a generalizable, flexible, and single-step statistical framework, *PhenMap*. This framework is an unsupervised machine-learning approach to simultaneously integrate cancer omics and phenome data to reveal clinically relevant and functional (discrete, when combined with a clustering method or continuous, by itself) subtypes with defined clinical indications. Here, we demonstrate *PhenMap* using two different examples and three different breast cancer datasets (n = 2045). Using *PhenMap* on one of the breast cancer datasets, we demonstrate the identification of subtypes that may predict response to a cyclin-dependent kinase (CDK)4/6 inhibitor. These biomarkers with further extensive validation may serve as biomarkers for the CDK4/6 inhibitors that are currently available for breast cancer patients with hormone receptor-positive metastatic/advanced breast cancer [11].

## 2. Results and Discussion

### 2.1. Overview of PhenMap

Here we propose a statistical framework—*PhenMap*—based on the machine-learning approach that merges with/without an unsupervised clustering method to delineate functional (discrete or continuous) subtypes (Figure 1; see Section 3). For efficient transcriptome-phenome integration and subtyping, the statistical model within *PhenMap* employs ‘hidden/mapping’ variables (termed context-specific mapping variables (CMVs), and are analogous to a set of gene networks having similar function) for integrative modeling of transcriptomics (or any other single omics) data with phenotypes (covariates). Unlike conventional clustering tools, CMVs are statistically and concurrently modeled together with quantitative and qualitative phenotypic covariates (including, but not limited to, drug responses and clinicopathological data) to identify those that are significantly associated with CMVs. This concurrent modeling of CMVs and covariates within the *PhenMap* framework reduces the loss of statistical information (see Section 3), unlike the conventional factor analysis method, which requires additional steps to model the associations between these parameters, leading to the loss of information. Furthermore, *PhenMap* involves a Bayesian implementation of noise reduction methods (sparsity-inducing priors; see Section 3), which allows for the optimal selection of significant features and phenotypes that capture the existing heterogeneity in the samples.

In most cases, the significant associations between CMVs and covariates may be sufficient to describe the transcriptome-phenome relationship and heterogeneity associated with samples, as continuous subtypes. However, for potential clinical applications, we further define “functional” and discrete subtypes using *PhenMap* by performing unsupervised clustering of CMVs alone or with corresponding significant covariates to identify functional subtypes (Figure 1; shown within a pale red box; list of all datasets used are in Appendix A). We demonstrate the utility of *PhenMap* below using two different examples of gene expression profiles from breast cancer (n = 2045). However, this method could be applied to any cancer types, diseases, and any type of single omics (transcriptome, methylome, etc.; single omics data at a time) data with sample-matched phenotypic information.

### 2.2. Example 1

#### 2.2.1. Example 1A—Identifying Functional and Discrete Subtypes in Breast Cancer with Drug Response Biomarkers

We first demonstrate the utility of *PhenMap* using transcriptome data (omics profile, after selecting highly variable <1000 genes) and the growth inhibition (GI_50_) values for four therapeutic compounds as phenotypes (covariates) from 37 breast cancer cell lines [12]. The drugs include etoposide (chemotherapy), fascaplysin (CDK4 inhibitor), bortezomib (proteasome inhibitor) and geldanamycin (heat-shock protein (HSP90) inhibitor; Appendix A; selection of the drugs is described in Appendix A). *PhenMap* analysis revealed two optimal CMVs with the highest Bayesian information criterion (BIC), which is a statistical method for model selection (see Section 3; Figure 2A and Appendix A). These CMVs were identified to be (concurrently and) significantly (*p* < 0.05; red dotted lines as a cut-off) associated with two drugs (fascaplysin and etoposide; out of four drugs; Figure 2B).

Furthermore, to improve the stratification of samples according to drug response, the two CMVs and GI_50_ values for fascaplysin and etoposide drugs (the two drugs significantly associated with the CMVs) were jointly clustered by consensus k-means clustering to identify discrete subtypes. This clustering analysis defined six “universal functional” (UF)-subtypes of samples; claudin-low, basal/inflammatory, luminal-HER2-1, -2, and -3, and UFs-6 (Appendix A and Figure 2C,D). Figure 2C shows UF-subtypes with large dots representing samples with increased sensitivity to etoposide, whereas Figure 2D shows the same with information from fascaplysin drug response. Since the number of samples within certain UF-subtypes was small, we applied a permutation-based approach to confirm the subtypes were robust. This approach revealed that five of six subtypes were robust (with average Euclidean distance > 1). The UFs-6 subtype was not robust as it had only two samples (Appendix A). Overall, our results showed at least five robust functional subtypes in breast cancer cell lines associated with drug responses that were identified using *PhenMap*.

Next, we sought to compare *PhenMap* to one of the widely used and conventional unsupervised methods—non-negative matrix factorization (NMF) [3,5,7]. By comparison, the NMF method produced only three subtypes (Appendix A). NMF subtypes were not able to distinguish luminal-Her2-1, -2, -3 and UFs-6 subtypes. These results suggest that *PhenMap* can be robust even with a small sample size of 37 cell lines and define functional (defined by selected drug information) transcriptome subtypes in breast cancer.

The six UF-subtypes were next compared to the intrinsic/etiological breast cancer subtypes defined in our previous study for these cell lines [12]. Claudin-low was the only intrinsic subtype retained in the UF-subtypes (Figure 2C,D) [12]. On the other hand, luminal and HER2 intrinsic subtype cell lines were sub-divided across three UF-subtypes into luminal-HER2-1, -2, and -3 (Figure 2C,D and Appendix A). This suggests further heterogeneity in luminal and HER2 subtypes that was previously reported by us and others [13,14]. For example, we recently demonstrated that luminal-A subtype alone can be stratified into at least five subtypes with differential cellular and immune characteristics and prognosis [13]. Similarly, the intrinsic basal breast cancer subtype cell lines combined with one HER2 line to form the UF basal/inflammatory subtype, as it was associated with the inflammatory colorectal cancer subtype [3] (Appendix A). The association of the single HER2 cell line with other basal breast cancer subtype may attribute to immune-enrichment or inflammatory characteristics. Previously, we reported that both basal and HER2 intrinsic subtypes are enriched for inflammatory characteristics, as assessed using our heterocellular signature comprising of different cell types of colorectal cancer [13]. Another subtype containing a basal subtype cell line and a HER2 line was termed UFs-6. Due to low sample size, the underlying characteristics of this subtype were not further possible to study. Overall, we identified six UF-subtypes using breast cancer cell lines that are different from intrinsic breast cancer subtypes.

Importantly, and unlike conventional methods, *PhenMap* clustered samples with quantitatively similar drug responses rather than simply by biological or intrinsic gene expression characteristics. For example, all the claudin-low and luminal-HER2-2 samples were moderately/highly sensitive to etoposide, and a majority was less sensitive (or resistant) to fascaplysin (Fisher and Kruskal-Wallis, *p* < 0.05; Figure 2E,F and Appendix A). By contrast, all of the luminal-HER2-3 subtype samples were moderately/highly sensitive to fascaplysin (Figure 2E,F and Appendix A), and seven out of eight basal/inflammatory lines were not highly sensitive to both drugs. All luminal-HER2-1 samples were resistant to both drugs, whereas all UFs-6 samples were highly sensitive (Figure 2E,F and Appendix A). These associations were not apparent when intrinsic subtypes, which were identified using conventional hierarchical clustering, were used (compare Appendix A and Figure 2E). Hence, the combined clustering of CMV and drug response provided a resolution to stratify cell lines according to their drug response and biological characteristics Overall, *PhenMap* identified unique drug response subtypes, particularly with respect to etoposide and fascaplysin sensitivity. The stratification of cell lines by fascaplysin sensitivity is potentially interesting, provided its similar mechanism of action to the CDK4/6 inhibitor palbociclib [15].

Next, we sought to further characterize a 576-gene signature that were simultaneously defined by *PhenMap* and were associated with the UF-subtypes (Figure 2G, Appendix A; top genes specific to each subtype shown in Appendix A). For example, *ZEB1*, known to be highly expressed in the claudin-low subtype, had a similar direction of association in the CMV space (with large positive values on loadings of CMV-1 and -2 in Figure 2G) as the claudin-low subtype (largest positive values on CMV-1 and -2 in Figure 2C,D). Similarly, we observed increased expression (and the same direction of associations on CMVs) of the basal and luminal markers *KRT17* and *ERBB3*, respectively (Figure 2C,D,G). Overall, *PhenMap* simultaneously provides statistically significant biomarkers and phenotypes associated with the subtypes, even with data of small sample size (n = 37).

For clinical validation of our UF-subtypes in patient-derived breast cancer samples (GSE42568 [16]), we developed prediction analysis of microarray (PAM) centroids [17], which represent the summarized expression of each UF-subtype using the 576-*PhenMap* gene signature (Appendix A). UFs-6 was removed due to its small sample size (n = 2). Five UF subtypes were found in 104 breast tumors with varying distributions (Figure 3A and Appendix A; see Appendix A). UF-subtypes were significantly associated with disease-free survival (DFS, Figure 3B), with the luminal-HER2-1 and basal/inflammatory subtypes, which were relatively resistant to both etoposide and fascaplysin in cell lines (Figure 2E,F), showing the worst DFS (Figure 3B). We also identified similar disease-specific survival associations in UF-subtypes in METABRIC breast cancer samples [18] (Figure 3A; Appendix A; n = 1904). Overall, *PhenMap* identified clinically relevant breast cancer subtypes potentially associated with chemotherapy and targeted therapy responses, and prognosis.

#### 2.2.2. Example 1B—Identifying “Context-Specific” Functional Subtypes in Breast Cancer Cell Lines

Since the CMVs from *PhenMap* are orthogonal to each other, we hypothesized that clustering individual CMVs with/without associated phenotypic covariates would provide CMV-specific subtypes that may be different from UF-subtypes and have various phenotypic associations (Figure 2B–D,G). We named these subtypes as “context-specific” functional subtypes. Hence, CMV-1 (from the 37 breast cancer training cell lines) was clustered with the GI_50_ values of the two drugs associated with it (Figure 2B). We identified six context-specific functional subtypes similar to the UF-subtypes (Appendix A).

In contrast, CMV-2, which was not significantly associated with any drug (Figure 2B), divided the cell lines into two CMV-2-specific subtypes (when clustered without the drug data; Figure 4A) such that one CMV-2 subtype was an unusual combination of claudin-low and luminal-HER2 breast cancer intrinsic subtypes. Although claudin-low lines are known to be quite different from luminal-HER2 lines, it is interesting that in this context, they combined to form a single subtype. As CMV-2 was not associated with any covariates (Figure 2B), significantly increased expression of genes such as *C3orf14*, *C8orf70*, *CMBL*, and *ANXA6* in claudin-low/luminal-HER2 subtypes (Appendix A) also explains the clustering together of these two intrinsic subtypes in CMV-2 subtypes.

The second CMV-2 subtype was almost entirely composed of the basal/inflammatory UF-subtype. The CMV-2-specific subtypes (PAM centroids in Appendix A) were also validated in tumors using 104 breast cancer samples (GSE42568 [16]; Appendix A). Interestingly, the two CMV-2 specific functional subtypes were prognostic (Figure 4B), with the basal/inflammatory subtype again displaying the worst DFS. Overall, this demonstrates the uniqueness and power of “context-specific” subtyping, which cannot be derived using other standard subtyping approaches.

While previous studies have reported claudin-low to be poor prognostic subtype, recently Fougner et al., demonstrated this interesting phenomenon that claudin-low subtype samples are present within different intrinsic subtypes of breast cancer with low proliferation and genomic instability [19]. Interestingly, they demonstrated that the prognosis of claudin-low subtype varies. Specifically, claudin-low subtype associated with luminal-A has good prognosis similar to ours in Figure 4B. Previously, we identified an association between stem cell signatures and luminal-A subtypes [13]. This suggests the heterogeneity and context-specific associations of the breast cancer subtypes.

### 2.3. Example 2—Identifying Continuous or Discrete Subtypes with Clinical Implications by Associating CMVs with Phenotypes Using PhenMap

Next, we applied *PhenMAP* to patient samples with clinical covariates as training data and evaluated how CMVs alone (that represent both low dimensional omics and abstract phenotypic information modeled together) can elucidate the omics-phenome relationship. In this example, *PhenMap* was directly applied to 101 patient-derived breast cancer expression profiles (GSE42568 [16]; 1000 most variable genes selected) with matched clinical phenotypes (age, tumor size, estrogen receptor (ER) status, nodal status, and grades (split into grade-1, -2, and -3); see Appendix A). Eight optimal CMVs were retained in this dataset (Appendix A); three were associated with at least one clinical phenotype (CMV-1, CMV-2, and CMV-3; Figure 5A) and five were prognostic (CMV-1, CMV-3, CMV-4, CMV-5, and CMV-7; Figure 5B). Interestingly, the significant association of CMV-1 with both ER negative (as scaled coefficient for ER+ is negative) and grade-3 (the highest grade; scaled coefficient is positive) statuses represent a subset of patients with poor prognosis in those patients (based on hazard ratio; HR; Figure 5A,B). On the other hand, CMV-2 is associated with ER negativity and also grade-1 (shown as negative scaled coefficient for grade-3 in CMV-2 in Figure 5A), but has no statistically significant change in prognosis in those patients (Figure 5A,B). These observations suggest that not all ER negative patients are associated with poor prognosis. Instead, other factors, such as young age and low grade (associated with CMV-2 in Figure 5A,B), may contribute to favorable patient prognosis, alongside with ER status. Similar observations have been indicated by others [18,20], however, these omics-phenome associations were not apparent in those studies based on conventional subtyping methods. Again, contradictorily, CMV-3 and its association only with grade-3 (but not with ER status), represent good prognosis (border line significance; Figure 5A,B). This suggests that a subset of patients with the highest grade could still have favorable survival irrespective of ER status. These observations need to be further validated.

Although CMVs-4 to -7 are not associated with any available phenotypes, CMV-4, -5, and -7 are still significantly associated with prognosis. While CMV-4 and -5 signify a poor prognosis, CMV-7 indicates a good prognosis (Figure 5A,B**)**. This suggests that there may be other factors (not available within this study) that may contribute to the association of prognosis with CMV-4, -5, and -7. This example suggests that patient prognosis is not just dependent on grade and ER statuses, but that the molecular features, together with the phenotypes within the CMVs, may play a role in patient survival. Overall, this further highlights the importance of jointly integrating single omics and phenome to understand how heterogeneity in cancer and other diseases are associated with clinical phenotype in a context-specific manner.

Further to explore this joint integration (and unlike in the first example), here, the CMVs were jointly clustered to define five general subtypes (G1–G5 subtypes; Figure 5C; Appendix A). Figure 5C,D show that the G2 and G3 subtypes were significantly associated with grade-3 (which are primarily ER-negative) tumors (Appendix A). Interestingly, we noticed that general subtypes were different from predicted UF-subtypes (in the first example) except for the enrichment of a G2-subtype with the basal/inflammatory UA-subtype (Figure 5E). This consistent basal association represents robustness of this subtype when clustered with other intrinsic subtypes. This analysis further highlights the importance of clustering CMVs with drug responses for subtype discovery (first example)—UF-subtypes provided a potential predictive classifier (for chemo/anti-CDK4/6 therapy), which warrants further validation. Overall, *PhenMap* can be applied to patient omics data with covariates to identify actionable subtypes.

## 3. Methods

### 3.1. PhenMap

Gene expression data for sample *i* with *p* genes (features), represented by *y*_i_, can be modeled within *PhenMap* framework using matched *c* covariates for the same sample, denoted by *x*_i_, where *y*_i_ = (*y*_i1_…*y*_i*p*_)^T^ and *x*_i_ = (*x*_i1_…*x*_i*c*+1_)^T^, T represents transpose of a matrix. This is carried out by assuming the existence of CMVs, *u*_i_ = (*u*_i1_…*u*_i*q*_)^T^, which captures the correlation structure in the expression data *y*_i_, where *q* is the number of CMVs. Hence, the model within *PhenMap* in Figure 1 can be written as follows,
*y*_i_ = **W***u*_i_ + *ξ*_i_,(1)
*u*_i_ = **β***x*_i_ + ε_i_,(2)
where **W** is a *p* × *q* projection matrix (loadings) relating each feature to the *q* CMVs, and **β** is a *q* × *c* matrix of regression coefficients quantifying the effect of covariates on the CMVs. The CMVs (*u*_i_), observed data errors (*ξ*_i_) and CMV errors (ε_i_) are assumed to be from a multivariate normal distribution (MVN), *u*_i_~MVN*_q_*[**β***x*_i_, **Φ**], *ξ*_i_~MVN*_p_* [0, **Σ**], and ε_i_~MVN*_q_* [0, **Φ**], respectively, where **Σ** = diag(*σ*_1_*^2^*… *σ_p_^2^*) and **Φ** = diag(*ϕ*_1_*^2^*… *ϕ_p_^2^*) are residual variances for both the observed data and CMVs, respectively. Bayesian methodology was employed to fit this model as it is known to perform better than deterministic algorithms when the data is of small sample size, a common scenario in biology.

The key assumption behind the model within *PhenMap* is that the observed correlations between features are due to CMVs such that conditional on these CMVs the features are independent of each other. Hence, the estimated loadings and regression coefficients (scaled by their corresponding standard deviations) can be used to identify features and phenotypes, respectively, associated with CMVs.

### 3.2. Sparseness and Prior Distributions Associated with Features and Phenotypes

In an effort to allow for automatic feature and phenotype selection, we introduced sparsity through priors on the elements of **W** and **β**, respectively. For features, the automatic relevance determination (ARD) prior [21], i.e., independent univariate Gaussian prior, is specified on each element *w*_jk_ of the loading matrix **W**, such that *p*(*w*_jk_|λ_jk_) = N[*w*_jk_|0, 1/λ_jk_], where *j* = 1…*p*, *k* = 1…*q* and λ_jk_ is a precision hyper-parameter which controls the contribution (loading) of feature *j* on the *k*^th^ CMV, *w*_jk_. The λ_jk_ has a gamma prior distribution *p*(λ_jk_|a, b) = G(λ_jk_|a, b). Large values of λ_jk_ result in shrinkage of *w*_jk_ towards zero, inducing sparsity in **W** (can be considered as statistically removing noisy features). The other variance parameters in **Σ** and **Φ** are also allowed to have independent gamma priors.

For regression coefficients **β**, *g*-priors [22] were adopted (as advantages), as they encourage some regularization of **β** and, also, parameter estimation using *g*-priors is invariant to changes in the scales of phenotypes [23]. An (*l* + 1)-dimensional multivariate Gaussian *g*-prior distribution centered at zero, with a parameter *g* to control the prior covariance, *p*(*β*_k_|$\phi_{k}^{2}$, *g*) = MVN*_l_*_+1_[*β*_k_|0, *g*(**X**^T^**X**)^−1^$\phi_{k\;}^{2}$], was considered on the rows of **β**. The role of *g* is to shrink the effect of non-informative phenotypes towards zero.

### 3.3. Model Selection

Model selection in *PhenMap* involves choosing the optimal number of CMVs representing the data. BIC [24] is a widely used method for model selection and is defined as BIC = 2*ll* – *C* log (*n*), where *ll* is the maximum log likelihood, *C* and *n* represent the number of parameters and samples, respectively. The highest BIC value indicates the best model. A regularized version of BIC was used in *PhenMap* models to select the optimal number of CMVs. This regularized BIC method avoids the disadvantage of using standard BIC, which performs poorly in high-dimensional data settings or when the model has multiple parameters. Additionally, this modification in BIC evaluates the likelihood of the maximum aposteriori (MAP) estimate instead of the maximum likelihood estimate [25,26]. However, this approach is computationally intensive. Hence, we also used a spike and slab prior [27] (a mixture of very peaky and broad Gaussians) over the loadings matrix, to allow for automatic selection of the number of CMVs. In addition, computational runtime profiles were evaluated for multiple versions of *PhenMap* (see Appendix A).

### 3.4. Model Convergence and Fitting in PhenMap

The Markov chain Monte Carlo (MCMC) algorithm allows us to sample from the target distribution (distribution of the *PhenMap* model). Trace plots (Appendix A), which are diagnostic plots showing the change in values of the algorithm at each MCMC iteration window, were used to assess the convergence of the chain (when the MCMC sampler is sampling from the distribution of the *PhenMap* model). Appendix A shows that the trace plots of the model parameters in *PhenMap* were mixing very well (samples for each parameter are relatively constant i.e., the colors are not mixing), and hence, were sampling from the target distribution (model convergence).

The fit of the model in *PhenMap* was assessed by comparing the differences between the distribution of the observed and the predicted (from the *PhenMap* model using the posterior predictive distribution [28]) breast cancer cell line (training) data. This assessment was performed by computing the mean absolute deviation (MAD) [29] between the covariance of the observed and each predicted data. Appendix A shows that majority of MAD values were less than one and close to zero, suggesting the *PhenMap* model fitted the data well.

### 3.5. The Algorithm in PhenMap

The posterior distribution of the model in *PhenMap* is complex and MCMC sampling techniques are required to produce samples from this complex distribution. The full conditional distributions of all the parameters exist in standard form. Hence, a Gibbs sampler [30] was constructed to iteratively sample from the target distribution. Here is a summary of the Gibbs sampler. For *s* = 1…*S*, the number of iterations or cycles;

(1)Derive the CMVs, **U**^(s)^, from a Gaussian full conditional distribution,(2)Derive the precision hyper-parameters **Λ**^(s)^ from a Gamma full conditional distribution,(3)Derive the loadings matrix **W**^(s)^ from a Gaussian full conditional distribution,(4)Derive the error covariance **Σ**^(s)^ from an inverse-Gamma full conditional distribution,(5)Derive the regression coefficients **β**^(s)^ from a Gaussian full conditional distribution, and(6)Derive the CMV covariance parameters **Φ**^(s)^ from an inverse-Gamma full conditional distribution.

### 3.6. Clustering of Context-Specific Phenotypic Mapping Variables (CMVs) with Drug Response Information

The CMVs and drug response information of the associated drugs (etoposide and fascaplysin) were jointly clustered using K-means consensus clustering. Firstly, the CMVs and the (growth inhibitory concentration) GI_50_ values (drug response information) were unit scaled for them to be comparable. ConsensusClusterPlus [31] *R* package was used to repeatedly (1000 times) cluster the data to eliminate the issue of dependence on initial conditions associated with K-means. The number of subtypes were varied from two to seven and both cophenetic coefficient [32] and silhouette width [33] measures were evaluated across subtypes to select the optimal number of universal functional (UF)-subtypes. Cophenetic coefficient measures cluster stability by evaluating the co-occurrences of samples within a cluster [32]. Silhouette width attempts to identify clusters with high between-class variability relative to within variability [33]. High values of both cophenetic coefficient and silhouette width indicate better clustering. Appendix A shows that good sample clustering was obtained when the number of UF-subtypes is six.

### 3.7. Development of Classifiers

In order to assign new samples into UF-subtypes and CMV-2-subtypes, we developed classifiers based on prediction analysis of microarray (PAM [17]) centroids for both 576 (Appendix A) and 179 (Appendix A) genes selected by *PhenMap*, respectively. PAM centroid represents scaled average expression of the signature in each subtype [17]. Using five-fold cross validation; PAM centroid with the least misclassification error rate (retaining all the genes selected by *PhenMap*) was generated for both classifiers. The expression pattern of a new sample was correlated to the PAM centroid and assigned to the subtype with highest Pearson correlation coefficients.

### 3.8. Datasets and Samples

Published gene expression data of breast cancer cell lines and drug response data from our original publication [12] were used as the training data to discover functional subtypes. The drugs included in this paper were selected based on their sensitivity to the breast cancer intrinsic subtypes, as shown in our previous study [12]. Hence, drugs were ordered by their corresponding false discovery rate-values and top and bottom 10 drugs based on sensitivity were selected. To ensure we maximized the number of cell lines (with complete drug response information) included in the training data, we selected 4 drugs with the maximum number of cell lines having matched gene expression data. Hence, the final training data consisted of 37 breast cancer cell lines with four drugs: etoposide, fascaplysin, bortezomib, and geldanamycin (Appendix A).

For validation of the UF-subtypes in tumors, GSE42568 [16] (n = 104) and METABRIC [18] breast cancer (n = 1904; download source: http://www.cbioportal.org/datasets; access date: 2017/05/10) gene expression data sets with their corresponding clinical information were used. Hence, in total, 2043 breast cancer samples were used in this work (Appendix A).

### 3.9. Availability of Data and Material

PhenMap resource: The PhenMap package is available as an R package on github (https://github.com/syspremed/PhenMAP/).

## 4. Conclusions

Our tool will be applicable clinically for stratifying cancers or other diseases provided large-scale data generated by The Cancer Genome Atlas (TCGA), International Cancer Genome Consortium (ICGC), and other similar consortia. Although this model can be applied to other individual- or multi-omics data types (including mutation, proteomics, radiomics, etc., with further modifications in the model), here, we used gene expression data that is widely available and clinically applicable to illustrate its potential. Currently, the model is limited to a single quantitative omics data type (containing continuous data) at a time. However, there can be multiple sample-matched phenotypic data of any types. Like any other factor-analytic model, *PhenMap* does not allow for joint modeling of survival time with omics data due to challenges presented while modeling censored information using high-dimensional feature matrix^16^. Nevertheless, the main contribution of this work is that it addresses the major challenges faced when discovering cancer subtypes, namely, to provide a meaningful biological interpretation and potential clinical utility to the discovered subtypes. UF subtypes provided a potential predictive classifier for chemotherapy and anti-CDK4 therapy warranting further validation (which is not within the scope of the current methodological study). Overall, *PhenMap* has the significant advantages over conventional clustering approaches of being able to simultaneously derive discrete or continuous functional subtypes, associate subtypes with phenotypes, provide biological/clinical implications, define subtype-specific biomarkers/signatures, and provide context-specific information.

## Figures and Tables

**Figure 1 cancers-12-02811-f001:**
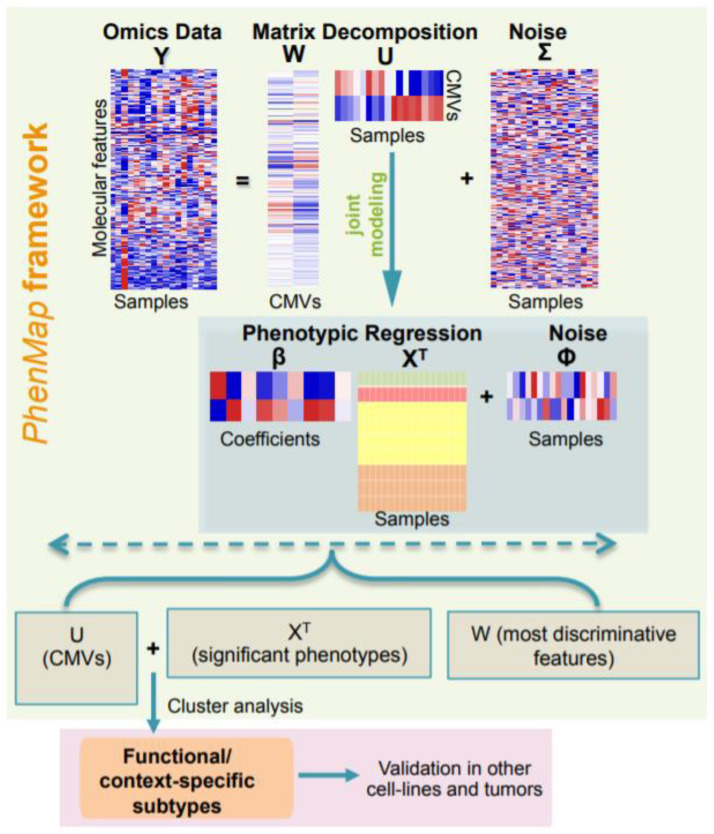
A flowchart of the steps involved in *PhenMap* framework. *PhenMap* is a machine-learning approach that maps the high-dimensional omics data matrix **Y** (with features or genes on the rows) to a low-dimensional matrix of context-specific mapping variables (CMVs), **U** (with samples on the columns) using a projection (loadings) matrix **W** (features on the rows). Matrix **Σ** (noise) represents part of the data that cannot be explained by the CMVs. The regression coefficient **β** estimates the effect of phenotypes in **X^T^** (samples on the columns) on the CMVs. Finally, it is optional that CMVs (**U**) and significantly associated phenotypes (**X^T^**) can be clustered to define discrete or continuous functional subtypes (shown as a separate box in magenta color).

**Figure 2 cancers-12-02811-f002:**
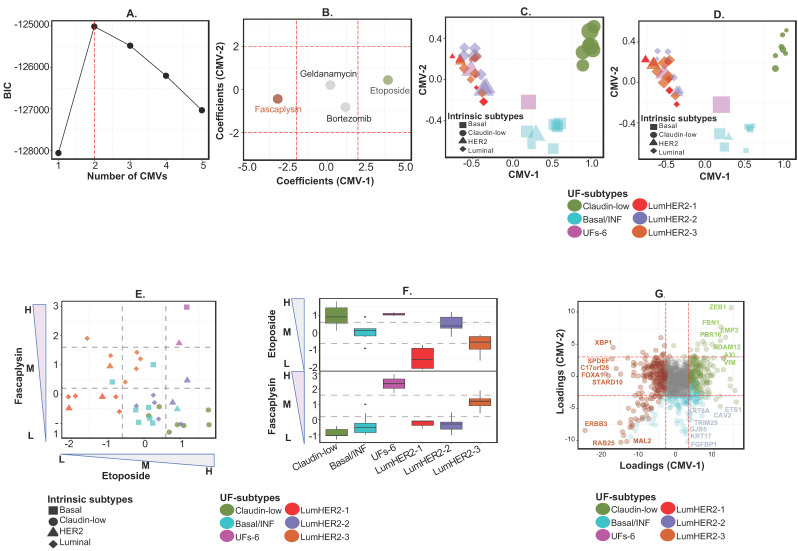
Identifying functional subtypes in breast cancer (BC) cell lines using *PhenMap* and transcriptome-phenome integration. (**A**) A Bayesian information criterion (BIC) plot to identify the optimal number of CMVs in 37 BC cell line gene expression dataset [12] (Y as in Figure 1). The red dashed line identifies the optimal number of CMVs (U as in Figure 1). (**B**) A plot showing the scaled regression coefficients (β as in Figure 1; the red dashed lines represent the (decimal point rounded) 5% significance level) for CMV-1 and 2. The red and green dots represent drugs (X^T^ as in Figure 1) with negative and positive significant effects on CMV-1, respectively. The grey dots represent drugs with non-significant effects on CMV-1 and -2. (**C**,**D**) Plots showing the results of clustering both the CMVs and GI_50_ significant drugs together to identify universal functional (UF)-subtypes. Six different colors identify the UF-subtypes, whereas intrinsic BC subtypes are denoted by different symbols. The larger the symbol sizes, the more sensitive the cell lines are to the drugs and vice versa—(**C**) etoposide and (**D**) fascaplysin. (**E**,**F**) Plots showing the scaled −log_10_ [GI_50_] values for etoposide and fascaplysin in this dataset, highlighting the sensitivity of the six UF-subtypes to the drugs. The grey dashed lines identify three response groups high (H), moderate (M), and low (L) sensitivity of the cell lines to the drug. The three response groups were determined by clustering the scaled -log_10_ [GI_50_] values of each drug seperately (see Appendix A). (**G**) The scaled loading coefficients (W as in Figure 1) represent significant genes associated with CMVs and UF-subtypes (the red dashed lines represent the 0.01% significance level and those in grey represent non-significant genes). The red dots denote genes with a negative effect on CMV-1 and up-regulated in all of the luminal-HER2 subtypes. On the other hand, the green and blue dots denote genes with a positive effect on CMV-1 and up-regulated in claudin-low and basal/inflammatory subtypes, respectively. INF—inflammatory.

**Figure 3 cancers-12-02811-f003:**
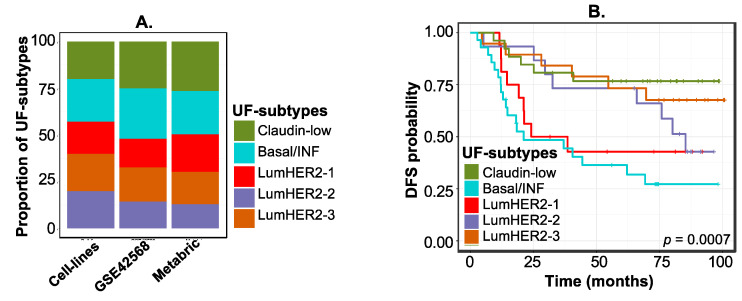
Validation of UF-subtypes and applying *PhenMap* to clinical samples. (**A**) A proportion plot of UF-subtypes in breast cancer (BC) cell lines and in breast tumors (from GSE42568 [15] and METABRIC [18]). (**B**) Kaplan-Meir plot for disease-free survival (DFS) of the predicted UF-Subtypes in GSE42568 [16] breast tumor data. *p* represents the log-rank test.

**Figure 4 cancers-12-02811-f004:**
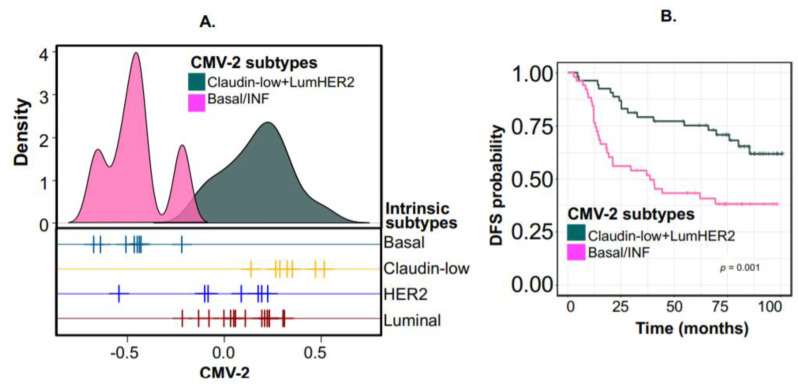
Defining context-specific subtypes. (**A**) A plot showing the clustering of breast cancer (BC) cell lines on CMV-2 into two context-specific functional subtypes (not UF-subtypes). The subtypes were compared to intrinsic BC subtypes. (**B**) Kaplan-Meier DFS plot of the two predicted CMV-2 functional subtypes in tumors (GSE42568 [15] data). *p* represents a log-rank test.

**Figure 5 cancers-12-02811-f005:**
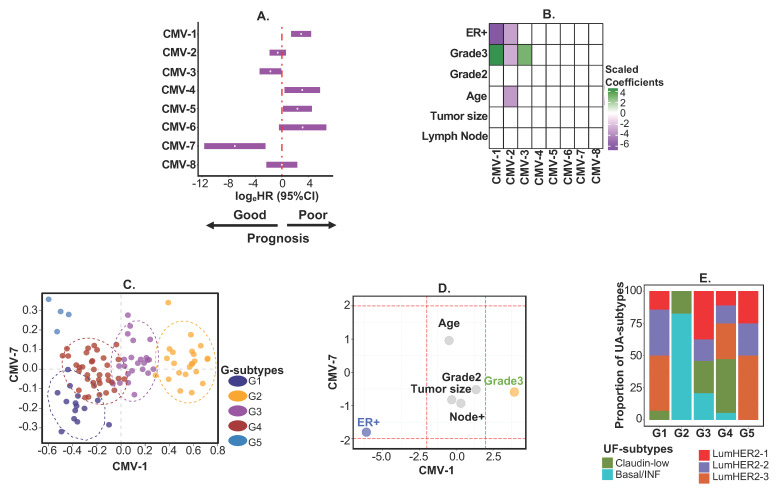
Application of *PhenMap* to clinical samples as a second example. (**A**) Forest plot assessing the prognostic value (DFS) of the eight CMVs selected by *PhenMap* applied to gene expression of GSE42568 [16] BC. Multivariate Cox regression estimates (log_e_(hazard ratio (HR))) and corresponding 95% confidence intervals (CI) for eight CMVs selected by *PhenMap* applied to breast cancer (BC) expression data (n = 101) from GSE42568 [16]. CMVs not including zero in their 95% CIs represent significant associations with DFS. (**B**) Heatmap of *PhenMap*-scaled regression coefficients for the effect of covariates on each CMVs. (**C**) CMV plot showing the clustering of the five identified general subtypes (G-subtypes) after applying *PhenMap* to gene expression of BC from GSE42568 [16] data. (**D**) Plot of the scaled regression coefficients for CMV-1 and CMV-7. The red dashed lines represent the (decimal point rounded) 5% significance level. (**E**) A proportion plot showing the association between G-subtypes and predicted UF-subtypes in the GSE42568 [16] data. For this analysis, age and tumor size were considered continuous variables, whereas the other parameters were considered categorical variables.

## Data Availability

Published gene expression data of breast cancer cell lines and drug response data from the original publication (Heiser and Sadanandam 2012) [20] were used. Other datasets include GSE42568 [22] (n = 104) and METABRIC [24] breast cancer (n = 1904; download source: http://www.cbioportal.org/datasets; access date: 10 May 2017).

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
