# Peer review of "A Machine-Learning Tool Concurrently Models Single Omics and Phenome Data for Functional Subtyping and Personalized Cancer Medicine"

_cancers, 2020, doi:10.3390/cancers12102811_

Round 1

Reviewer 1 Report

The authors introduced a novel genome-phenome machine-learning integration tool to clarify the functional and phenotype-integrated subtypes of cancers. The new tool could integrate omics and phenome data. The study was well-organized. However, several revisions are needed as listed below.

  1. I think that the manuscript would improve if the authors could explain the methods and results in detail. 
  2. The authors showed that claudin-low breast cancer was not associated with poor survival (Figure 3B). Those results were different from previous reports (Dias, et al. PLoS ONE, 2017).  The authors should explain the reason for it.

Reviewer 2 Report

This manuscript describes a tool, PhenMap, for unsupervised clustering of samples. I do not have too much problem with the algorithm itself; however in my opinion some of the terminologies are quite misleading. The authors should also provide some more evidences that this algorithm performs better than existing methods. Below please find my reviews.

  1. I was originally very interested in this work due to its title “causal-learning;” however upon checking the methods and results I was a bit disappointed by the fact that this work has nothing to do with the causality of the phenotypes. Indeed the authors extracted context-specific phenotype mapping variables, or CMVs, from the genotypes; however you cannot say that this has anything to do with “causal-learning” or “causal-mapping.” The reason is that you are still looking for correlations instead of causality. I suggest the authors check the articles discussing causality (e.g. Nature Machine Intelligence 2:369-375, 2020) if they want to really pursue this route.

  1. Another problem with the title is that the authors claimed to have integrated omics and phenome data. Firstly there is nothing new with this approach; people working on machine learning or statistical inference need to do that such “integration.” Secondly this title misled me into believing that the authors have done something into the “integration” process, say multi-omics data integration.

  1. This title also hinted that the authors have done something to eliminate the batch effect; however I do not see any methodology and discussion related to batch effect elimination process.

  1. While the CMV methodology looks interesting, it looks really to me is another dimensional reduction method. And there are already quite some popular numbers of dimensional reduction methods, among them PCA, MDS, to name just a few. I wonder how and whether this method performs compared to PCA, say whether the first two principle components can form a better or worse clustering of the samples (Figure 2C and 2D), etc. I would love to see that the CMV indeed extracted meaningful information from the genotype data and can form better clusters.

  1. The guidance in setting the number(s) of CMVs to be extracted based on BIC looks ok in the first glance; however it also means that we need to try various CMV numbers and compare their BICs in order to know the optimal number of CMVs. Although for simplicity’s sake it is useful to just set the number of CMVs as 2 in order to perform PCA-like analysis, in reality the users need to go through all sorts of CMV numbers and compare their BICs. I wonder if the authors have some better ways to help users selecting optimal CMVs (or implement an automatic pipeline for best CMV number).

  1. Also regarding to the number of CMVs. Since the authors only go through very small number of CMVs (say, 1-5, as indicated in Figure 2A), I wonder if the BIC is going to go up again for some larger number of CMVs. If so then restricting the number of CMVs to very small number may be misleading.

  1. Github repository is inaccessible (404 not found). Perhaps there is a typo in the https address? Please also indicate the github repo in the abstract.

Round 2

Reviewer 2 Report

The manuscript looks fine to me now. I have no further questions.